# Novel methotrexate long-acting system using ambroxol coating and hydroxypropyl methylcellulose encapsulation for preferential and enhanced lung cancer efficiency

Samaa Abdullah[1]*, Najla Altwaijry[2], Meaad Alnakhli[3], Ghezlan ALenezi[3], Alaa A. Al-Masud[4], Hanan Henidi[5], Yahya F. Jamous[6]

1 College of Pharmacy, Amman Arab University, Amman, Jordan, 2 Department of Pharmaceutical Sciences, College of Pharmacy, Princess Nourah Bint Abdulrahman University, Riyadh, Saudi Arabia, 3 College of Pharmacy, Princess Nourah Bint Abdulrahman University, Riyadh, Saudi Arabia, 4 Tissue Banking Section, Research Department, Health Science Research Center, Princess Nourah Bint Abdulrahman University, Riyadh, Saudi Arabia, 5 Research Department, Natural and Health Sciences Research Center, Princess Nourah Bint Abdulrahman University, Riyadh, Saudi Arabia, 6 The National Centre of Vaccines and Bioprocessing, King Abdulaziz City for Science and Technology (KACST), Riyadh, Saudi Arabia

* s.abdullah@aau.edu.jo

**Data Availability Statement:** All data are in the manuscript and/or supporting information files.

## Abstract

Methotrexate (MTX) is classified as an antimetabolite. It's commonly used to treat lung cancer. MTX is an immunosuppressant following the above-mentioned mechanism of action due to its poor selectivity. The tricky move is to investigate the solid dispersions and coating using a co-delivery system of MTX and Ambroxol (ABL). ABL is known for its, anticancer and preferential pulmonary distribution after oral administration. The goals was development were the MTX physiochemical modulation for pulmonary enhanced distribution, MTX resistance modulation and long-acting system development using ABL middle coating and HPMC outer coating. The selection of the optimum MTX-ABL dispersion was done based on the FT-IR characterization. The MTX-release analysis results for the optimized MTX-ABL solid dispersion and the HPMC-coated MTX-ABL gel product were tested for release in the gastrointestinal simulated media to select the most optimum HPMC amounts to coat the MTX-ABL optimum solid dispersion. Moreover, different characterizations of FT-IR, X-ray diffraction and Scanning electron micros-copy investigations were completed for the MTX, ABL, the ABL-MTX optimized solid disper-sion and the optimum MTX-ABL-HPMC gel. The cytotoxicity assay and the ELISA to assess the levels of BAX, BCL-2, TGF-β and FR-α after the MTX, ABL and the optimized MTX-ABL solid dispersion groups were tested against lung cancer cells, A549 cells, for 24 h. The sus-tained release character and HPMC-ABL encapsulation of MTX were confirmed. The MTX-ABL solid dispersion showed less MTX resistance without the need to use the high MTX con-centrations in comparison to the MTX alone. The apoptotic, anti-metastatic, and MTX prefer-ential lung cancer uptake profiles were higher using the MTX-ABL solid dispersion than in the MTX or ABL. The MTX-ABL-HPMC gel could serve as an alternative to the MTX-oral tablets available in the markets with enhanced efficacy and safety profile.

**Funding:** Princess Nourah bint Abdulrahman University Researchers Supporting Project number (PNURSP2024R89).

**Competing interests:** The authors have declared that no competing interests exist.

## Introduction

Public health is facing an increasing threat from cancer, with 17 million new cases reported in 2018. Lung cancer is one of the leading causes of cancer-related deaths worldwide, affecting people of both genders. Small cell lung cancer (SCLC) and non-small cell lung cancer (NSCLC) are the two main subtypes that are often distinguished. About 85% of lung cancer cases are of the predominant type, NSCLC, with SCLC accounting for the remaining 15%. Estimates indicate a significant rise in cancer diagnoses, with 27.5 million new cases each year by 2040 [1].

Methotrexate (MTX) is a drug used to treat several ailments, including multiple sclerosis, psoriasis, and several cancers, including lung, bone, breast, and cervical cancer. However, its poor solubility in water (0.01 mg/mL at 20˚C) and limited bioavailability (18% at doses>40 mg/m^2) limit its effectiveness. One of the ways that methotrexate is used to treat lung cancer is through its anti-metabolic properties, which prevent DNA, RNA, and protein synthesis. MTX mostly targets and inhibits dihydrofolate reductase (DHFR), an enzyme required for the synthesis of tetrahydrofolate, an intermediate in the synthesis of purines and pyrimidines. MTX inhibits DHFR, which prevents the synthesis of nucleotides required for DNA replication and cell division, thereby reducing the development of cancer cells. Among the issues connected to MTX use are its adverse effects on the oral and gastric mucosa [2–5].

The tricky move is to investigate the solid dispersions and coating using a co-delivery system of MTX and ambroxol (ABL). ABL is known for its mucolytic, anticancer, antiviral, immune modulation activities and preferential pulmonary distribution after oral administration. The ABL's high pulmonary perfusion can add to the MTX-ABL solid dispersion perfusion after the oral administration to target lung cancer using MTX to decrease the MTX systemic adverse effects [6–9]. Interestingly, ABL was proven to boost the phagocytosis of lung cancer-dead cells after the chemotherapeutic treatment in an *in vivo* trial as published before [10]. The goals of this solid dispersion development were the MTX physiochemical modulation for pulmonary enhanced distribution, MTX resistance modulation and long-acting system development using ABL middle coating and Hydroxypropyl methylcellulose (HPMC) outer coating. The HPMC outer layer was known for its use in the development of sustained-release systems [11]. The apoptotic profile of the lung cancer cells could be investigated using the measurement of the protein expression of the anti-apoptotic B-cell lymphoma-2 protein (BCL-2), and pro-apoptotic Bcl-2-associated protein x (BAX) [12]. The MTX-developed cancer resistance and metastasis profile could be investigated using the measurement of the protein expression of Transforming Growth Factor-β (TGF-β) [13]. The enhanced lung cancer cells' uptake of the MTX could be investigated using the measurement of the protein expression of Folate Receptor-ɑ (FR-ɑ). Moreover, one of the aims was to lower the MTX dose needed due to the use of ABL's anticancer actions using different mechanisms. This product can serve as a strong alternative to the MTX-oral tablets available in the markets with enhanced efficacy and safety profile.

## Materials and methods

### Materials

Hydroxypropyl methylcellulose (HPMC) and methotrexate (MTX) were used; they were sourced from Sigma-Aldrich in the United States. The source of Ambroxol (ABL)'s donation was Dar-Aldawaa, Jordan. The Modified Eagle Medium (DMEM) from Dulbecco (Gibbco, USA). We bought 10% fetal bovine serum (FBS) from Gibbco in the United States. Trypsin-EDTA and phosphate-buffered saline (PBS) were obtained from Thermo-Fisher, USA. We

bought the MTT cell proliferation test reagent from Invitrogen in the United States. The work was approved by the institutional review board at Princess Nourah bint Abdulrahman University, Saudi Arabia under the number HAP-01-R-059.

## MTX method of analysis

Using the chemical substance's capacity to absorb UV light, UV-Spectroscopy (Thermo-scientific, USA) is a quantitative technique used to measure the quantity of MTX and ABL. The MTX was diluted and prepared in various quantities using distilled water (pH 7.4). Five separate concentrations of the MTX-calibration curve in the range of 0.004–1.024 mg/mL for MTX standard solutions were measured to assess the linearity of the procedure. To determine each concentration's matching absorbance, measurements were made at 372 nm [2, 14].

## Solid dispersion preparation and selection of the optimum MTX-ABL ratio

MTX solution and ABL solution were made in equivalent quantities and ratios of varied values, as indicated in **Table 1**. This step was introduced to cover the MTX using ABL. In 5 mL of water, MTX or ABL powder was dissolved to create both solutions. The experimental settings were standardized for reliable comparisons by keeping this constant concentration. Based on the desired proportions, calculations were made to modify the ratio between MTX and ABL. 200 mg of MTX and 400 mg of ABL were dissolved in 5 mL of water for a 1:2 ratio of MTX to ABL. Likewise, in the case of a 1:3 ratio, 200 mg of MTX would require 600 mg of ABL to preserve the designated ratio. Following this reasoning, the corresponding ABL quantity would be 800 mg to maintain the 1:4 ratio of 200 mg of MTX dissolved in 5 mL of water. This illustrates the careful consideration given to preserving precise ratios in the experimental design for accurate and controlled outcomes. For a night, the mixtures were allowed to dry under the fume hood. A more uniform and smaller particle size distribution of powder size of less than 250 μm might be achieved by grinding the dry MTX-ABL solid dispersion with a mortar and pestle [15].

The selection of the optimum MTX-ABL solid dispersion ratio was based on the Fourier transform infrared spectra (FT-IR) of the different combinations. FT-IR (Agilent Technology, USA) was used to illustrate and interpret the interaction between the ingredients of MTX, ABL, and MTX-ABL solid dispersion using different ratios. Each sample was compressed using the stainless-steel pin of the instrument. The sample was scanned at a laser frequency of 15799 cm$^{-1}$ at medium resolution [6].

## Assay of the optimum solid dispersion

This study used to determine the assay of MTX was performed in acetone. MTX-ABL optimum solid dispersion (20 mg) was added to 15 mL of acetone. The sample was subjected to vortexing for 5 min. When the sample was centrifuged, the absorbance for the supernatant layer was determined by UV spectrophotometry at 372 nm in triplicates after dilution with water [14].

**Table 1. Combination ratios to prepare different MTX-ABL ratios.**

| MTX-ABL Ratio | ABL amount (mg) | MTX amount (mg) |
|:---:|:---:|:---:|
| 1: 1 | 200 | 200 |
| 1: 2 | 400 | |
| 1: 3 | 600 | |
| 1: 4 | 800 | |

**Table 2. A demonstration of the various formulations' optimization components utilizing a 30 mg/mL HPMC gel.**

| Combination. No. | HPMC amount (mg) | MTX: ABL ratio (mg: mg) | MTX: ABL: HPMC ratio | Gel total volume (mL) |
|---|---|---|---|---|
| 1 | 200 | 100: 400 | 1: 4: 2 | 7 |
| 2 | 400 | | 1: 4: 4 | 14 |
| 3 | 600 | | 1: 4: 6 | 21 |

## MTX-ABL solid dispersion HPMC encapsulation and optimization

Table 2 displays the combinations that were evaluated. These combinations involved combining varying amounts of 30 mg/mL of HPMC gel with a set MTX-ABL optimal solid dispersion quantity. The combinations were then assessed for their release patterns in contrast to the MTX-ABL optimum solid dispersion. For the optimization, the dialysis bag experiment in 0.1N HCl (pH 1.2) for two hours, followed by two hours in pH 6.8 for the various formulations in Table 2 and the MTX-ABL optimal solid dispersion, served as the basis for the MTX release investigation. 300 millilitres of either the simulated intestinal fluid (SIF) with a pH of 6.8 or the simulated gastric fluid (SGF) with a pH of 1.2 were used to submerge the study groups of comparable 100 mg MTX at a controlled temperature of 37±0.05 ˚C. Additionally, every hour, the release material was completely replaced. MTX release was measured with UV spectroscopy at 372 nm [14].

## Characterizations of the optimum MTX-ABL solid dispersion and the selected MTX-ABL-HPMC gel

**Fourier Transforming-Infrared (FT-IR).** FT-IR (Agilent Technology, USA) was used to illustrate and interpret the results of MTX, ABL, HPMC, optimum MTX-ABL solid dispersion, selected MTX-ABL-HPMC gel, and ABL-HPMC blank gel. Each sample was compressed using the stainless-steel pin of the instrument. The sample was scanned at a laser frequency of 15798.7 $cm^{-1}$ at medium resolution.

**Powder X-ray Diffraction (PXRD).** A Rigaku, Ultima IV, Japan equipment was used to evaluate the MTX, ABL, HPMC, optimal MTX-ABL solid dispersion, chosen MTX-ABL-HPMC gel, and ABL-HPMC blank gel. Cu-Kb reduction with nickel filter at 40 kV and 40 mA produced X-rays. The scan range (2θ) encompassed 5 to 70 degrees at a speed of 10 degrees per minute [16].

**Scanning Electron Microscopy (SEM).** SEM (JEOL, Japan) is a highly versatile technique used to obtain high-resolution images (2D) and detailed surface information. The morphology and distribution of MTX, ABL, HPMC, optimum MTX-ABL solid dispersion, selected MTX-ABL-HPMC gel, and ABL-HPMC blank gel were examined by SEM using an electron microscope and a sputter coater The dried specimens were mounted on a metal stub (with double-sided adhesive tape), coated in a vacuum with silver and platinum, and scanned at an accelerating 30 kV voltage [17].

**Effects on lung cancer cytotoxicity assay.** Dulbecco's Modified Eagle Medium (DMEM) supplemented with 10% fetal bovine serum (FBS) was used to cultivate A549 lung cancer cells and streptomycin-penicillin to guard against bacterial infection in cell culture until 90–95% confluency is reached in an incubator set at 37 ˚C with 5% CO2. The process of subculturing involved twice washing the cell monolayers in 4 mL of phosphate-buffered saline (PBS) from Thermo-Fisher, USA, and then adding 3 mL of trypsin-EDTA from Thermo-Fisher, USA [1]. The vitality and proliferation of the cell growth were assessed using the MTT cell proliferation assay kit (Invitrogen, USA). The MTT test is regarded as a calorimetric cytotoxicity indicator. In a humidified 5% CO2 incubator, A549 cells were plated in 96-well plates (5 × 10^3 cells/

well) and incubated for 24 hours at 37 ˚C. MTX, ABL, optimal MTX-ABL solid, and control were the study groups. The groups' corresponding MTX treatment concentrations were 20, 40, 60, 80, 100, and 120 μg/mL. These values may be related to the ABL concentrations of 80, 160, 240, 320, 400, and 480 μg/mL, in that order. The treatment duration was 24 h. To solubilize the formazan, the culture supernatant was withdrawn after 24 hours and replaced with 100 μL dimethyl sulfoxide (DMSO). The plate was incubated for four hours at 37 ˚C in a humidified 5% $CO_2$ incubator. After that, a microplate reader was used to measure the absorbance at 570 nm. Three duplicates of the experiment were conducted [1].

**Effects on lung cancer apoptosis, resistance and MTX uptake.** For the A549 lung cancer cells' levels of the anti-apoptotic B-cell lymphoma-2 protein (BCL-2), pro-apoptotic Bcl-2-associated protein x (BAX), Transforming Growth Factor-β (TGF-β) and Folate Receptor-α (FR-α), commercially available ELISA kits (Invitrogen, USA) were used, MTX-uptake, resistance and apoptotic markers were estimated as per the manufacturer's instruction after the applying the treatment groups of MTX, ABL, optimal MTX-ABL solid using the 50%-Inhibitory Concentration (IC50) of MTX in comparison to the control [18–20].

## Statistical interpretation

Tukey's Multiple Comparison Test was used after one-way ANOVA for statistical analysis, and the results were expressed as mean±standard deviations (SD). In the statistical analysis, which was carried out with Graph Pad Prism 4.0 software (Graph Pad Software San Diego, USA), $p < 0.05$ was considered statistically significant.

## Results and discussion

### MTX method of analysis

The linearity of the method (**Fig 1**) was determined by measuring five independent concentrations of the MTX-calibration curve in the range of 0.004–1.024 mg/mL for MTX standard solutions. Each concentration was measured at 372 nm to have its corresponding absorbance. The beer's Lambert equation was A = 0.5791*C (mg/mL) + 0.6785. The absorptivity/slope was 0.5791 mg/mL, and the y-intercept was 0.6785 [2, 14].

### Solid dispersion selection of the optimum MTX-ABL ratio and assay

MTX solution and ABL solution were made in equivalent quantities and ratios of varied values, as indicated in **Table 1**. This step was introduced to cover the MTX using ABL [15]. FT-IR spectra were conducted for MTX, ABL, and MTX-ABL solid dispersions with different ratios as illustrated in **Fig 2**. For the MTX spectrum, a characteristic absorptions band as a broad signal for O-H stretching of carboxyl groups overlapping with the OH stretching from crystallization water at 3500 cm^-1 and (N-H stretching) at 3000 cm^-1. In addition, the sharp peak at 1690–1450 cm^-1 corresponds to C-O stretching from the carboxylic group [3].

For ABL spectrum, (-OH) at 3400 cm^-1, (-NH2) 2600 cm^-1, (-NH) between 3200 and 3300 cm^-1, (C = C aromatic) between 1700 and 1600 cm^-1, (C-H aliphatic) between 2905 and 2740 cm^-1, (C-O stretch) at 1000 cm^-1 and (C-Br stretch) between 710 and 620 cm^-1 [6].

For the solid dispersions' spectra with different ratios, the spectra of MTX-ABL 1:1, 1:2, and 1:3 ratios indicate the presence of MTX coating on ABL. FT-IR characteristics were utilized to identify the surface layer of the examined material [21]. The purpose of the ABL solid dispersion was to provide a covering layer for MTX with preferential lung absorption due to the ABL presence, aiming to lower the systemic side effects and adjust the cancer cells' uptake and

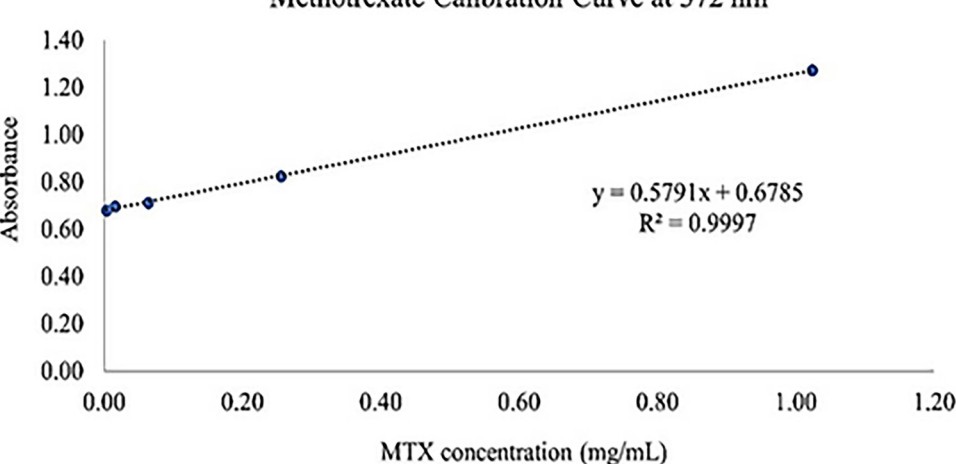

**Fig 1. UV-spectroscopy calibration curve for MTX.**

resistance [9, 22]. Consequently, the initial ratios related to MTX-ABL solid dispersions were excluded, and the final spectrum of MTX-ABL solid dispersion ratio of 1: 4 was adopted. The solid dispersion spectrum of the MTX-ABL 1:4 ratio exhibits a peak similar to ABL, with a prominent peak at 3500–2500 cm^-1, indicating the formation of a hydrogen bond between the hydroxyl groups of ABL and the primary amine MTX. This achievement could fulfil one objective of the study [6, 21, 23]. For the assay of the optimum MTX-ABL solid dispersion (1: 4 ratio), each 20 mg of the solid dispersion of MTX-ABL comprised 4.49±0.29 mg of MTX.

## MTX-ABL solid dispersion HPMC encapsulation and optimization

**Table 2** displays the combinations that were evaluated. These combinations involved combining varying amounts of 30 mg/mL of HPMC gel with a set MTX-ABL optimal solid dispersion quantity. The combinations were then assessed for their release patterns in contrast to the MTX-ABL optimum solid dispersion.

For the optimization, the release experiment shown in **Fig 3** demonstrates that the lowest release rates and percentages at the end of the SGF and SIF stages were using the MTX-ABL-HPMC (1: 4: 4) gel. The MTX-ABL-HPMC (1: 4: 4) gel release rate and end percentages in SGF were similar to the MTX-ABL (1: 4) solid dispersion (Similarity Factor = 87.72), but the MTX-ABL (1: 4) solid dispersion SIF end release percentages was higher than the MTX-ABL-HPMC (1: 4: 4) gel by 48%. This indicates the necessity to encapsulate the solid dispersion with the HPMC matrix to control the solid dispersion swelling and dissolution. On the other hand, the MTX-ABL-HPMC (1: 4: 4) gel release end percentages in SGF and SIF were lower than the MTX-ABL-HPMC (1: 4: 2) gel by 20% and 15%, respectively. This could indicate the HPMC amount needed to fully encapsulate, encounter, and accomplish the needed interactions with the MTX-ABL (1:4) solid dispersion. Moreover, the MTX-ABL-HPMC (1: 4: 4) gel release end percentages in SGF were lower than the MTX-ABL-HPMC (1: 4: 6) gel by 5%, but the MTX-ABL-HPMC (1: 4: 4) gel release end percentage in SIF was the same of MTX-ABL-HPMC (1: 4: 6) gel. However, the SIF release rate of the MTX-ABL-HPMC (1: 4: 4) gel was lower than the MTX-ABL-HPMC (1: 4: 6). This could lead to the optimum HPMC amount needed to be adsorbed on the surfaces of the MTX-ABL (1:4) solid dispersion to hinder the swelling and dissolution of MTX [16]. The most optimum MTX-ABL-HPMC gel was the MTX-ABL-HPMC (1: 4: 4) gel. Interestingly, the release of the

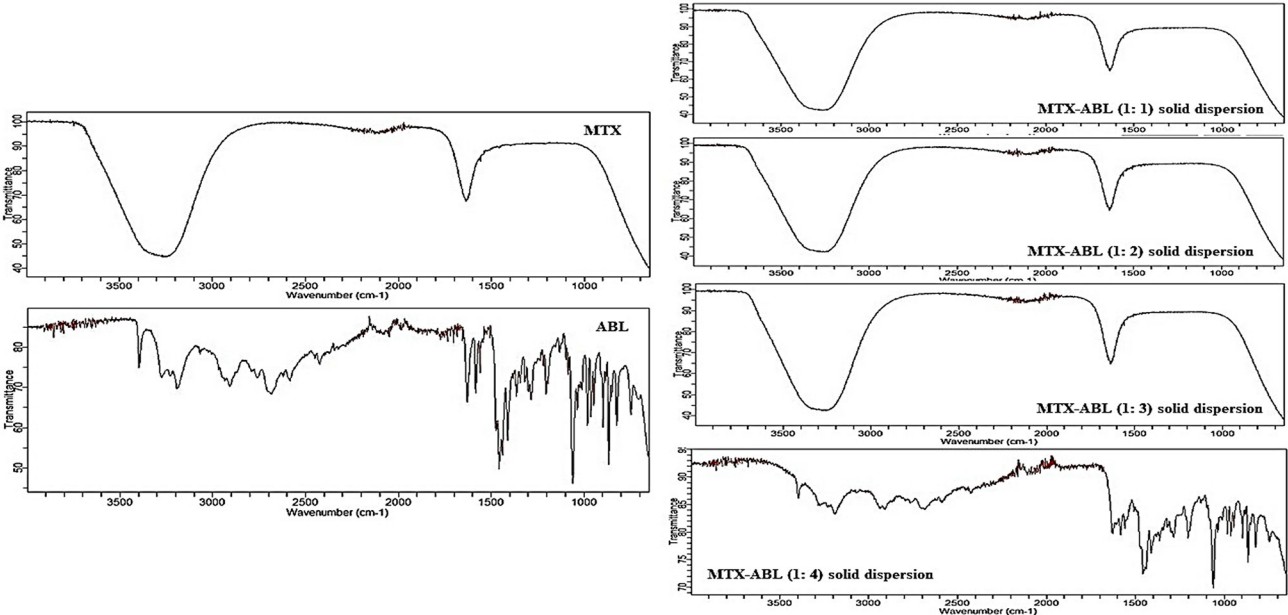

**Fig 2. FT-IR spectra of MTX, ABL and different ratios of MTX-ABL solid dispersions.**

MTX-ABL-HPMC (1: 4: 4) gel was higher in the SIF than the SGF to target the area of MTX-ABL particle absorption at the duodenum.

## Characterizations of the optimum MTX-ABL solid dispersion and the selected MTX-ABL-HPMC gel

**Fourier Transforming-Infrared (FT-IR).**   The discussion of MTX and ABL spectra in previous could be used to understand the interactions introduced in **Fig 4**. The HPMC spectrum came in consistent with the literature [11]. However, the spectrum of MTX-ABL-HPMC (1: 4: 4) gel shows a similar spectrum of the HPMC alone with diminished intensities, especially at the range of 3500–1800 cm^-1. This could affirm the HPMC surface coating, and hydrogen bond formation between the oxygen of the HPMC's ether with the hydrogen hydroxyl and/or amine of the ABL. This was affirmed by the spectra of ABL-HPMC (1: 1) blank gel. The spectrum of ABL-HPMC (1: 1) blank gel shows a similar spectrum of the HPMC alone with diminished intensities, especially at the range of 3500–1800 cm^-1. This could affirm the HPMC surface coating, and hydrogen bond formation between the oxygen of the HPMC's ether with the hydrogen hydroxyl and/or amine of the ABL. After all, the FT-IR results suggest that the MTX core particles were coated in the middle layer of the ABL, and the HPMC was the outer layer of the gel to cover the MTX after oral administration. This could elicit the MTX systematic safety and selective pulmonary absorption by the HPMC outer and ABL middle layers, respectively [7, 11, 15].

**Powder X-ray Diffraction (PXRD).**   For **Fig 5**, the MTX's diffractogram shows definite peaks in the range of 8–40 degrees to prove crystalline nature. The ABL's diffractogram shows a more noisy profile than MTX's one in the range of 6–45 degrees to indicate ABL's amorphous characteristics. The diffractogram of MTX-ABL (1: 4) solid dispersion presents a crystalline diffractogram with similarity to the ABL's due to the coating of the MTX, and with lower intensities than the ABL's diffractogram alone due to the dilution effects of MTX. Moreover, the HPMC's diffractogram shows a predominant noisy and amorphous characteristic in

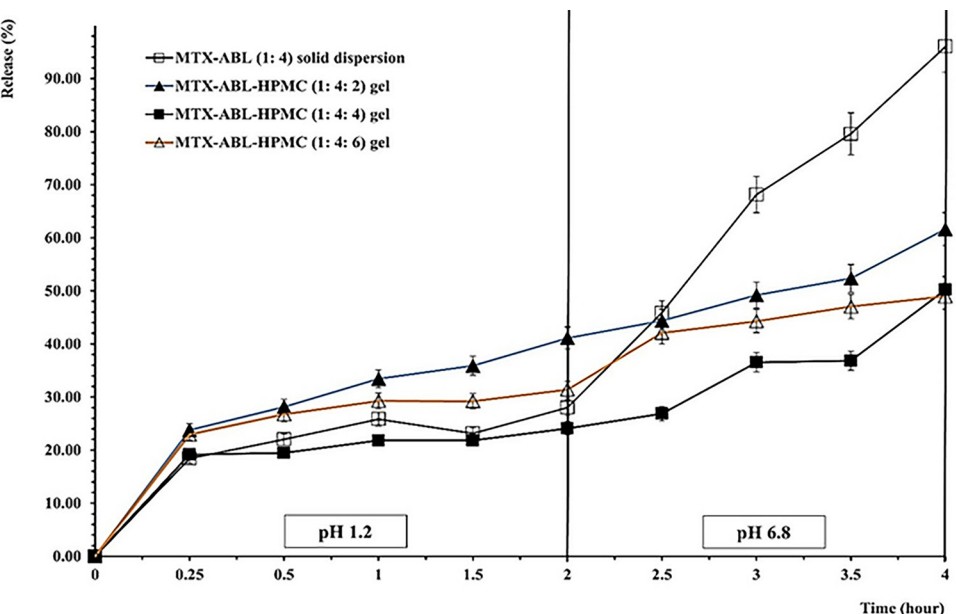

**Fig 3. MTX-release using dialysis bag experiment for the MTX-ABL (1: 4) solid dispersion, and different HPMC ratios of the MTX-ABL-HPMC gel for optimization.**

the range of 6–65 degrees. Interestingly, the diffractogram of MTX-ABL-HPMC (1: 4: 4) gel shows a combination of ABL's and HPMC's diffractograms to approve their coating with enhanced crystallinity in comparison to the HPMC and ABL diffractograms. The blank gel of

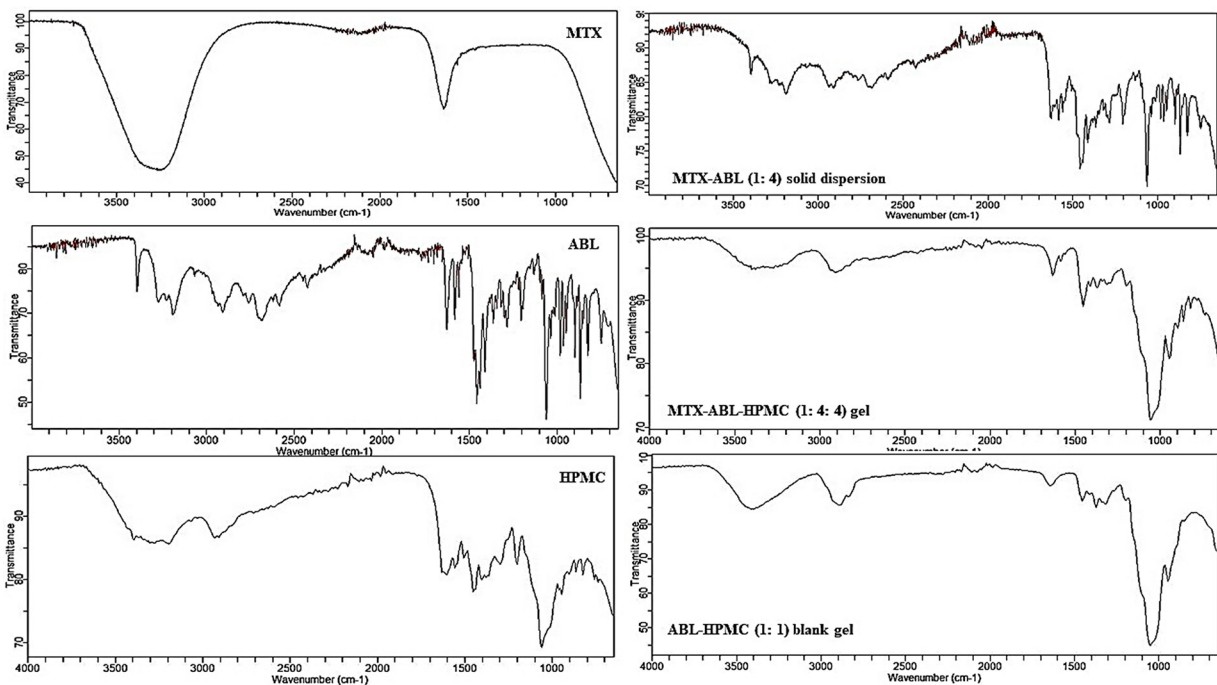

**Fig 4. FT-IR spectra of MTX, ABL, HPMC, MTX-ABL (1: 4) solid dispersion, the MTX-ABL-HPMC (1: 4: 4) gel and ABL-HPMC (1: 1) blank gel.**

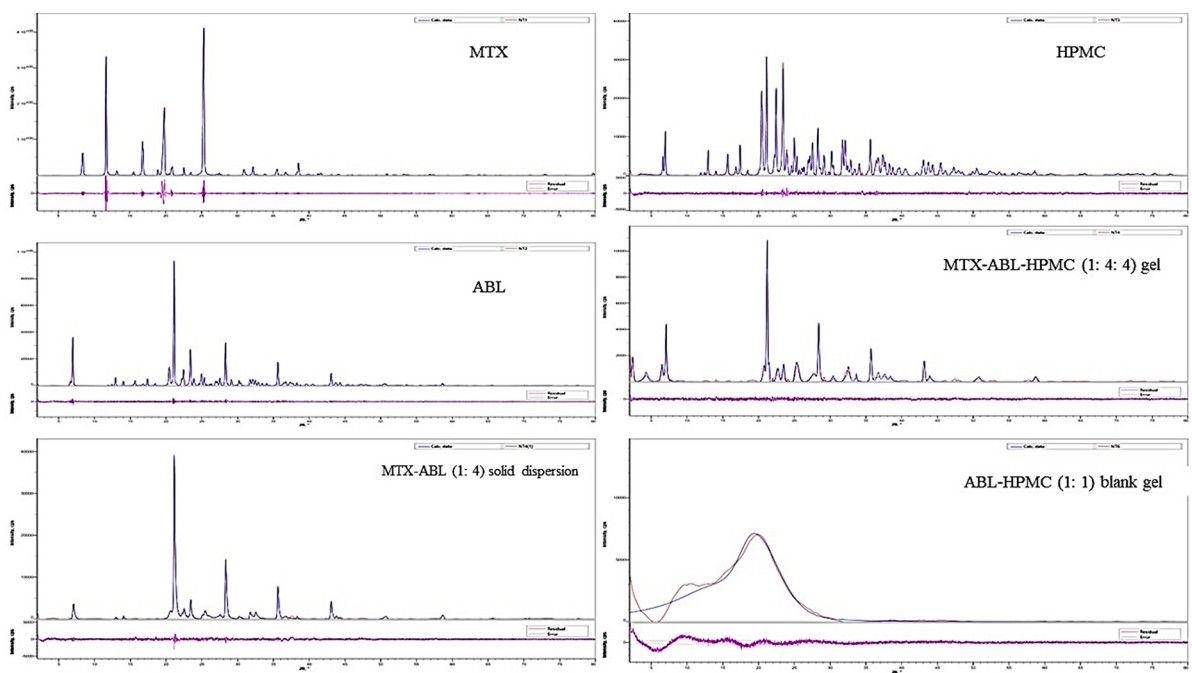

**Fig 5. PXRD diffractogram of MTX, ABL, HPMC, MTX-ABL (1: 4) solid dispersion, the MTX-ABL-HPMC (1: 4: 4) gel and ABL-HPMC (1: 1) blank gel.**

the ABL-HPMC (1: 1) diffractogram indicates a unique large peak in the range of 5–30 degrees to indicate the amorphicity in comparison to the MTX-ABL-HPMC (1: 4: 4) gel diffractogram.

**Scanning Electron Microscopy (SEM).** For Fig 6, the MTX's image shows the large crystalline nature. The ABL's image shows the combination of crystals and irregular-shaped particles with lower particle sizes than the MTX crystals. The image of MTX-ABL (1: 4) solid dispersion presents a similar powder shape and size to the ABL's image that could approve the size reduction of MTX after the ABL coating. Moreover, the HPMC's image shows a non-uniform larger particle than the ABL, but smaller than MTX's size. The MTX-ABL-HPMC (1: 4: 4) gel's image, which was prepared as films, shows white small particles of MTX that were missed in the image of the blank gel of ABL-HPMC (1: 1) film. This could indicate that the MTX was fully encapsulated by the gel of ABL-HPMC [17].

**Effects on lung cancer cytotoxicity assay.** The cell viability results of different MTX, ABL, and optimum MTX-ABL (1: 4) solid dispersion treatment concentrations (Fig 7) show a decrease in the cell viability by increasing the MTX dose up to 40 μg/mL with IC50 25.21 ±2.05 μg/mL, but it shows an increase in the cell viability by increasing the MTX dose more than 40 μg/mL due to the formation of cell resistance [24]. As the dose of MTX increases due to the MTX resistance, side effects will increase. In comparison, the ABL shows less effectiveness than MTX with IC50 of 161.32±3.44 μg/mL. The MTX-ABL (1: 4) solid dispersion shows a significant increase in the anti-cancer activity in comparison to MTX or ABL alone with IC50 of 121.64±4.11 μg/mL for ABL and 30.41±2.31 μg/mL for MTX and p-value< 0.05 in the range of 40–100 μg/mL with a decrease in the MTX resistance, and increasing the efficacy of ABL. The IC50 of MTX (25.21 μg/mL), ABL (161.32 μg/mL) and MTX-ABL solid dispersion (121.64 μg/mL for ABL and 30.41 μg/mL for MTX). The MTX-ABL (1: 4) anticancer activity in the range of 40–100 μg/mL of MTX concentrations was superior to the MTX and ABL alone, which could be correlated to the resolved MTX resistance in the presence of solid

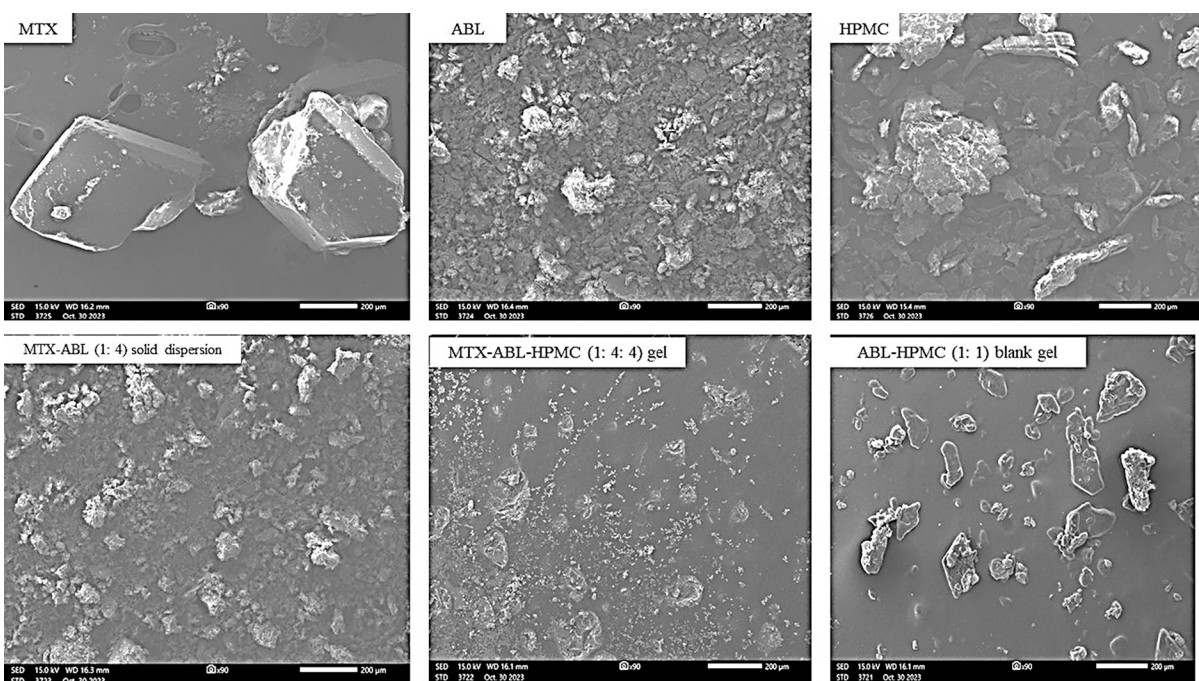

**Fig 6. SEM image of MTX, ABL, HPMC, MTX-ABL (1: 4) solid dispersion, the MTX-ABL-HPMC (1: 4: 4) gel and ABL-HPMC (1: 1) blank gel at 90x magnification (200 μm scale).**

dispersion, to avoid the need to increase the MTX dose. In addition, the ABL anticancer activity results of the MTX-ABL solid dispersion were better than the ABL alone anticancer results in the range of 20–100 μg/mL [3].

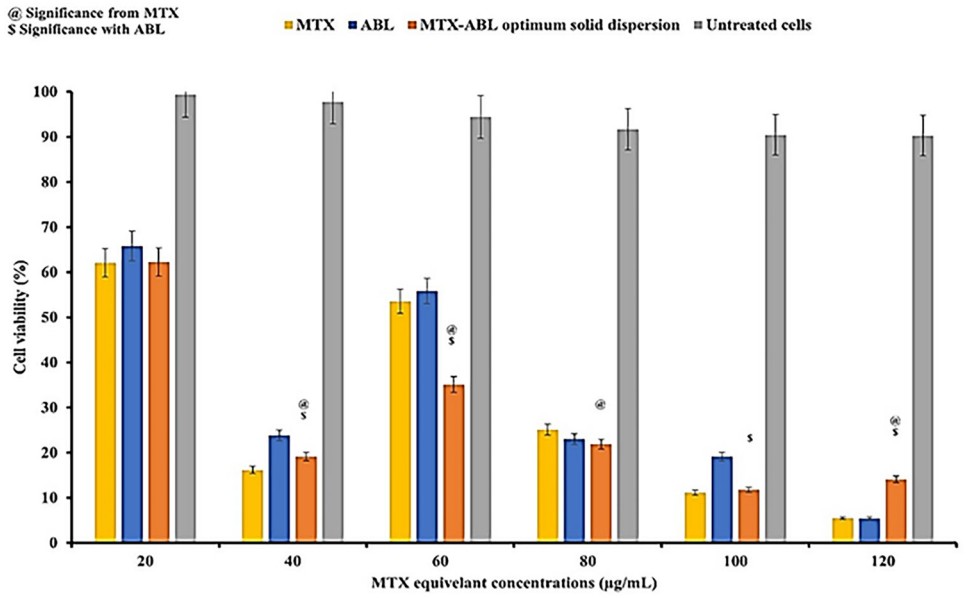

**Fig 7. MTT assay against lung cancer cells of MTX, ABL, and MTX-ABL (1: 4) solid dispersion.**

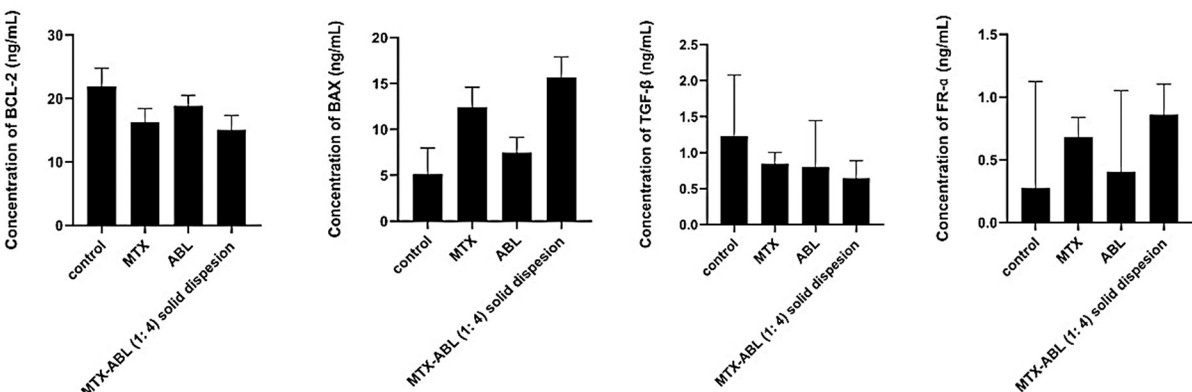

**Fig 8. BCL-2, BAX, TGF-β and FR-α expression in lung cancer cells after MTX, ABL, and MTX-ABL (1: 4) solid dispersion treatment.**

### Effects on lung cancer apoptosis, resistance and MTX uptake

Based on the MTT results of the solid dispersions, the treatment dose of IC50 of 121.64 ±4.11 μg/mL for ABL and 30.41±2.31 μg/mL for MTX were used and reflected to generate **Fig 8**. For the apoptotic activity, BCL-2 expression level after the MTX-ABL (1: 4) solid dispersion was less than the control, MTX and ABL groups by 5.11, 2.01 and 4.31 ng/mL, respectively. On the other hand, BAX expression level after the MTX-ABL (1: 4) solid dispersion was more than the control, MTX and ABL groups by 10.03, 3.24 and 9.11 ng/mL, respectively. As a result, the lung cancer apoptotic profile of the MTX-ABL (1: 4) solid dispersion was superior to the other groups, which could indicate the pharmacological modulation and synergism implemented after the ABL dispersion and coating of the MTX (**Fig 8**) [12].

For the resistance and metastatic profiles, TGF-β expression level after the MTX-ABL (1: 4) solid dispersion was less than the control, MTX and ABL groups by 1.05, 0.6 and 0.25 ng/mL, respectively. This could indicate, with the support of the cytotoxicity data, that the ABL coating and raw material have implicated lower cancer cell resistance and metastasis tendency (**Fig 8**) [13].

For the MTX uptake, FR-α expression level after the MTX-ABL (1: 4) solid dispersion was more than the control, MTX and ABL groups by 1.07, 0.32 and 0.84 ng/mL, respectively. As a result, the MTX lung cancer uptake of the MTX-ABL (1: 4) solid dispersion was superior to the other groups which could indicate the enhanced lung cancer uptake. The FR-α is highly expressed in the A549 lung cancer cells and has been involved in the MTX mechanism of action (**Fig 8**) [22, 25].

## Conclusion

MTX is classified as an antimetabolite. It's commonly used to treat lung cancer. MTX is an immunosuppressant following the above-mentioned mechanism of action due to its poor selectivity. The tricky move is to investigate the solid dispersions and coating using a co-delivery system of MTX and ABL. ABL is known for its mucolytic, anticancer, antiviral, immune modulation activities and preferential pulmonary distribution after oral administration. The goals was development were the MTX physiochemical modulation for pulmonary enhanced distribution, MTX resistance modulation and long-acting system development using ABL middle coating and HPMC outer coating. The selection of the optimum MTX-ABL dispersion was done based on the FT-IR characterization. The MTX-release analysis for the optimized MTX-ABL solid dispersion and the HPMC-coated MTX-ABL gel product was tested for

release in the gastrointestinal simulated media to select the most optimum HPMC amounts to coat the MTX-ABL optimum solid dispersion. Moreover, different characterizations of FT-IR, X-ray diffraction and Scanning electron microscopy investigations were completed for the MTX, ABL, the ABL-MTX optimized solid dispersion and the optimum MTX-ABL-HPMC gel. The cytotoxicity assay of the MTX, ABL and the optimized MTX-ABL solid dispersion were tested against lung cancer cells, A549 cells, for 24 hours using the MTT-cell viability assay. After all, the protein expression analysis after the MTX, ABL and the optimized MTX-ABL solid dispersion were tested against lung cancer cells for 24 hours using the Enzyme-Linked Immunosorbent Assay (ELISA) to assess the levels of BAX, BCL-2, TGF-β and FR-α. The sustained release character and HPMC-ABL encapsulation of MTX were confirmed. In the MTT assay, the MTX-ABL solid dispersion showed less MTX resistance without the need to use the high MTX concentrations in comparison to the MTX alone. However, the apoptotic, anti-metastatic, and MTX preferential lung cancer uptake profiles were higher using the MTX-ABL solid dispersion than in the MTX or ABL. Moreover, one of the aims was to lower the MTX dose needed due to the use of ABL's anticancer actions. The developed system was designed to elicit the selectivity profile as it was proven on the cellular level after 24 hours of treatment to increase the expression level of Folate Receptor-α (FR-α). The MTX lung cancer uptake of the MTX-ABL (1: 4) solid dispersion was superior to the other groups which could indicate the enhanced lung cancer uptake. The FR-α is highly expressed in the A549 lung cancer cells and has been involved in the MTX mechanism of action. However, a long-term *in vivo* absorption study will be recommended to solidify the claim and identify the resistance profile developed. Moreover, the molecular level and signalling investigations to identify how ABL can elicit the BAX expression in the cancer cells will be the next step. Interestingly, ABL was proven to boost the phagocytosis of lung cancer-dead cells after the chemotherapeutic treatment in an *in vivo* trial as published before [10]. This could encourage that the data in the *in vivo* trial is worthy of investigation. After all, this product can serve as a strong alternative to the MTX-oral tablets available in the markets with enhanced efficacy and safety profile.

## Supporting information

**S1 Fig.**
(JPG)

**S2 Fig.**
(JPG)

**S3 Fig.**
(JPG)

**S4 Fig.**
(JPG)

**S5 Fig.**
(JPG)

**S1 Data.**
(XLSX)

**S2 Data.**
(XLSX)

**S3 Data.**
(RAR)

**S4 Data.**
(RAR)

**S1 Code.**
(DOCX)

## Acknowledgments

Part of this work was presented at the **Controlled Release Society (CRS) Annual Meeting and Exposition, in July 2024**.

## Author Contributions

**Conceptualization:** Samaa Abdullah, Najla Altwaijry.

**Data curation:** Samaa Abdullah, Ghezlan ALenezi, Alaa A. Al-Masud, Hanan Henidi.

**Formal analysis:** Samaa Abdullah, Najla Altwaijry, Ghezlan ALenezi, Alaa A. Al-Masud, Hanan Henidi.

**Funding acquisition:** Najla Altwaijry, Alaa A. Al-Masud, Yahya F. Jamous.

**Investigation:** Samaa Abdullah, Najla Altwaijry, Alaa A. Al-Masud.

**Methodology:** Samaa Abdullah, Meaad Alnakhli, Ghezlan ALenezi, Hanan Henidi.

**Project administration:** Samaa Abdullah, Alaa A. Al-Masud.

**Resources:** Najla Altwaijry, Meaad Alnakhli, Yahya F. Jamous.

**Software:** Samaa Abdullah, Ghezlan ALenezi, Yahya F. Jamous.

**Supervision:** Samaa Abdullah.

**Validation:** Samaa Abdullah, Yahya F. Jamous.

**Visualization:** Samaa Abdullah, Meaad Alnakhli, Ghezlan ALenezi.

**Writing – original draft:** Samaa Abdullah, Najla Altwaijry.

**Writing – review & editing:** Meaad Alnakhli, Alaa A. Al-Masud, Hanan Henidi.

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
