## [Decision Letter · Decision Letter 0]

4 Sep 2024

PONE-D-24-31096Novel methotrexate long-acting system using ambroxol coating and hydroxypropyl methyl cellulose encapsulation for preferential and enhanced lung cancer efficiencyPLOS ONE

Dear Dr. Abdullah,

Thank you for submitting your manuscript to PLOS ONE. After careful consideration, we feel that it has merit but does not fully meet PLOS ONE’s publication criteria as it currently stands. Therefore, we invite you to submit a revised version of the manuscript that addresses the points raised during the review process.

**ACADEMIC EDITOR: **

We look forward to receiving your revised manuscript.

Kind regards,

Muhammad Muzamil Khan

Academic Editor

PLOS ONE

Journal Requirements:

"Princess Nourah bint Abdulrahman University Researchers Supporting Project number (PNURSP2024R89)"

5. We note that the original protocol that you have uploaded as a Supporting Information file contains an institutional logo. As this logo is likely copyrighted, we ask that you please remove it from this file and upload an updated version upon resubmission.

Reviewers' comments:

Reviewer's Responses to Questions

**Comments to the Author**

1. Is the manuscript technically sound, and do the data support the conclusions?

Reviewer #1: Yes

2. Has the statistical analysis been performed appropriately and rigorously? 

Reviewer #1: Yes

3. Have the authors made all data underlying the findings in their manuscript fully available?

Reviewer #1: Yes

4. Is the manuscript presented in an intelligible fashion and written in standard English?

Reviewer #1: No

5. Review Comments to the Author

Reviewer #1: To identify potential negative points regarding the research study described in the manuscript, consider the following aspects:

1. The study mentions that methotrexate (MTX) has poor selectivity as an immunosuppressant, which is a fundamental challenge.

2. The study does not provide long-term resistance data, which could be a limitation.

3. Ambroxol, despite its anticancer properties, is traditionally used as a mucolytic agent, and its long-term effects when repurposed for cancer therapy are not fully understood.

4. The study is primarily focused on in vitro tests only using A549 lung cancer cells!

5. The manuscript needs extensive revision for language and grammar. Some editing for the English language is required throughout the manuscript due to too many mistakes.

6. The quality of some of the figures should be improved.

6. PLOS authors have the option to publish the peer review history of their article (what does this mean?). If published, this will include your full peer review and any attached files.

Reviewer #1: No

---

## [Author Response · Author response to Decision Letter 0]

23 Oct 2024

Pointwise Response to the Reviewers (Answer in RED):

ACADEMIC EDITOR: 

Answer: Thank you. It was uploaded.

Answer: Thank you. It was uploaded.

Answer: Thank you. It was uploaded.

• Answer: Thank you. It was included in the cover letter.

Answer: Thank you. It was reflected in the manuscript.

"Princess Nourah bint Abdulrahman University Researchers Supporting Project number (PNURSP2024R89)"

Answer: Thank you. It was included in the cover letter and amended as the following.

“The authors extend their appreciation for the financial support of Princess Nourah bint Abdulrahman University Researchers Supporting Project number (PNURSP2024R89), Princess Nourah bint Abdulrahman University, Riyadh, Saudi Arabia”

 Answer: Thank you. It was added to the “Materials” section.

4. When completing the data availability statement of the submission form, you indicated that you would make your data available on acceptance. We strongly recommend all authors decide on a data sharing plan before acceptance, as the process can be lengthy and hold up publication timelines. Please note that, though access restrictions are acceptable now, your entire data will need to be made freely accessible if your manuscript is accepted for publication. This policy applies to all data except where public deposition would breach compliance with the protocol approved by your research ethics board. If you are unable to adhere to our open data policy, please kindly revise your statement to explain your reasoning and we will seek the editor's input on an exemption. Please be assured that, once you have provided your new statement, the assessment of your exemption will not hold up the peer review process.

 Answer: Thank you. It was included in amended as the follows.

“All data are in the manuscript and/or supporting information files”.

5. We note that the original protocol that you have uploaded as a Supporting Information file contains an institutional logo. As this logo is likely copyrighted, we ask that you please remove it from this file and upload an updated version upon resubmission.

Answer: Thank you. It was uploaded after checking with the IRB issuing institution that it is in the most updated form.

Answer: Thank you. It was added as the following at the end of the manuscript.

“Supporting Information files

• The institutional review board at Princess Nourah bint Abdulrahman University, Saudi Arabia under the number HAP-01-R-059 was attached.

• The code system included in the experimnets.

• Raw results of calibration curve, Assay, FT-IR, SEM, PXRD, release, MTT and ELISA readings.”

Reviewers' comments:

Reviewer's Responses to Questions

Comments to the Author

1. Is the manuscript technically sound, and do the data support the conclusions?

Reviewer #1: Yes

Answer: Thank you.

2. Has the statistical analysis been performed appropriately and rigorously? 

Reviewer #1: Yes

Answer: Thank you.

3. Have the authors made all data underlying the findings in their manuscript fully available?

Reviewer #1: Yes

Answer: Thank you.

4. Is the manuscript presented in an intelligible fashion and written in standard English?

Reviewer #1: No

Answer: The authors would like to extend their appreciation to the reviewer. A professional language editing service was requested and applied in the revised version of the manuscript.

5. Review Comments to the Author

Reviewer #1: To identify potential negative points regarding the research study described in the manuscript, consider the following aspects:

1. The study mentions that methotrexate (MTX) has poor selectivity as an immunosuppressant, which is a fundamental challenge.

Answer: The authors would like to appraise their appreciation for the reviewer’s point. The developed system was designed to elicit the selectivity profile as it was proven on the cellular level after 24 hours of treatment to increase the expression level of Folate Receptor-ɑ (FR-ɑ). The MTX lung cancer uptake of the MTX-ABL (1: 4) solid dispersion was superior to the other groups which could indicate the enhanced lung cancer uptake. The FR-ɑ is highly expressed in the A549 lung cancer cells and has been involved in the MTX mechanism of action. However, a long-term in vivo absorption study will be recommended to solidify the claim (ADDED to the Conclusions).

2. The study does not provide long-term resistance data, which could be a limitation.

Answer: The authors would like to confirm that the long-term in vivo absorption study will be recommended to solidify the claim and identify the resistance profile developed (ADDED to the Conclusions).

3. Ambroxol, despite its anticancer properties, is traditionally used as a mucolytic agent, and its long-term effects when repurposed for cancer therapy are not fully understood.

Answer: ABL was tested by the first author in previous articles on lung cancer repurposing (Md et al., 2021). We can affirm that the authors will take the reviewer's insight into the next investigation on the molecular level and signalling to identify how ABL can elicit the BAX expression in cancer cells. Interestingly, ABL was proven to boost the phagocytosis of lung cancer dead cells after the chemotherapeutic treatment in an in vivo trial as published before (Zhang et al., 2017). This could encourage that the data in the in vivo trial is worthy of investigation (ADDED to the Conclusions).

4. The study is primarily focused on in vitro tests only using A549 lung cancer cells!

Answer: The authors would like to confirm that the long-term in vivo absorption study will be recommended to solidify the claim and identify the resistance profile developed (ADDED to the Conclusions).

5. The manuscript needs extensive revision for language and grammar. Some editing for the English language is required throughout the manuscript due to too many mistakes.

Answer: The authors would like to extend their appreciation to the reviewer. A professional language editing service was requested and applied in the revised version of the manuscript (ADDED to the Conclusions).

6. The quality of some of the figures should be improved.

Answer:

Thank you. The authors have enhanced the figure quality to support the claim given. Kindly note that the submission system file decreases the Figures' quality in the PDF. Authors are kindly advised to download the figures separately (ADDED to the Conclusions).

References

Md, S., Abdullah, S. T., Alhakamy, N. A., Bani-Jaber, A., Radhakrishnan, A. K., Karim, S., . . . Rizg, W. Y. (2021). Ambroxol Hydrochloride Loaded Gastro-Retentive Nanosuspension Gels Potentiate Anticancer Activity in Lung Cancer (A549) Cells. Gels, 7(4). doi:10.3390/gels7040243

Zhang, X., Chen, Q., Chen, M., Ren, X., Wang, X., Qian, J., . . . Sha, X. (2017). Ambroxol enhances anti-cancer effect of microtubule-stabilizing drug to lung carcinoma through blocking autophagic flux in lysosome-dependent way. Am J Cancer Res, 7(12), 2406-2421.

---

## [Decision Letter · Decision Letter 1]

19 Nov 2024

Novel methotrexate long-acting system using ambroxol coating and hydroxypropyl methyl cellulose encapsulation for preferential and enhanced lung cancer efficiency

PONE-D-24-31096R1

Dear Dr. Samaa Abdullah

We’re pleased to inform you that your manuscript has been judged scientifically suitable for publication and will be formally accepted for publication once it meets all outstanding technical requirements.

Kind regards,

Muhammad Muzamil Khan

Academic Editor

PLOS ONE

Additional Editor Comments (optional):

Reviewers' comments:

Reviewer's Responses to Questions

**Comments to the Author**

1. If the authors have adequately addressed your comments raised in a previous round of review and you feel that this manuscript is now acceptable for publication, you may indicate that here to bypass the “Comments to the Author” section, enter your conflict of interest statement in the “Confidential to Editor” section, and submit your "Accept" recommendation.

Reviewer #2: All comments have been addressed

2. Is the manuscript technically sound, and do the data support the conclusions?

Reviewer #2: Yes

3. Has the statistical analysis been performed appropriately and rigorously? 

Reviewer #2: Yes

4. Have the authors made all data underlying the findings in their manuscript fully available?

Reviewer #2: Yes

5. Is the manuscript presented in an intelligible fashion and written in standard English?

Reviewer #2: No

6. Review Comments to the Author

Reviewer #2: Abstract:

Problem 1:

You have mentioned in 2nd line “MTX is an immunosuppressant following the above-mentioned mechanism of action due 24 to its poor selectivity” but there is no mechanism of action you have discussed/wrote earlier.

Assay of the optimum solid dispersion

Problem 2:

In the method you have both vortexed and centrifuged the MTX and ABL, kindly explain how both these methods helped you? Please also explain how much dilution was made with water before taking UV spectrum?

MTX-ABL solid dispersion HPMC encapsulation and optimization

Problem 3:

In line 132 please explain, the meaning of sentence, as it is not getting clarified. “For the optimization, the dialysis bag experiment in 0.1N HCl (pH 1.2) for two hours, followed by two hours in pH 6.8 for the various formulations in Table 2 and the MTX-ABL optimal solid dispersion, served as the basis for the MTX release investigation.

Results and Discission

Solid dispersion selection of the optimum MTX-ABL ratio and assay

Problem 4:

FTIR Spectrum should be in the same figure using suitable software rather than placing each spectrum individually. It will make the comparison easy of all functional groups.

Problem 5:

No evident peaks of MTX-ABL (ratio 1:4) was observed in the solid dispersion formulation in Figure 2

Characterizations of the optimum 258 MTX-ABL solid dispersion and the selected MTX-ABL-HPMC gel Fourier Transforming-Infrared (FT-IR)

Problem 6:

Please use suitable and relevant software for the comparison in FTIR as requested earlier.

Effects on lung cancer apoptosis, resistance and MTX uptake

Problem 7:

How have you calculated the dose?

Problem 8:

Few grammatical and typographical error were observed throughout the manuscript. Carefully read the manuscript and rectify those errors.

7. PLOS authors have the option to publish the peer review history of their article (what does this mean?). If published, this will include your full peer review and any attached files.

Reviewer #2: No

---

## [Editor Report · Acceptance letter]

26 Nov 2024

PONE-D-24-31096R1 

PLOS ONE

Dear Dr. Abdullah, 

I'm pleased to inform you that your manuscript has been deemed suitable for publication in PLOS ONE. Congratulations! Your manuscript is now being handed over to our production team.

Kind regards, 

on behalf of

Dr. Muhammad Muzamil Khan 

Academic Editor

PLOS ONE